# Continual Unsupervised Representation Learning

**Dushyant Rao, Francesco Visin, Andrei A. Rusu,**
**Yee Whye Teh, Razvan Pascanu, Raia Hadsell**[*]
DeepMind
London, UK

## Abstract

Continual learning aims to improve the ability of modern learning systems to deal with non-stationary distributions, typically by attempting to learn a series of tasks sequentially. Prior art in the field has largely considered supervised or reinforcement learning tasks, and often assumes full knowledge of task labels and boundaries. In this work, we propose an approach (CURL) to tackle a more general problem that we will refer to as *unsupervised continual learning*. The focus is on learning representations without any knowledge about task identity, and we explore scenarios when there are abrupt changes between tasks, smooth transitions from one task to another, or even when the data is shuffled. The proposed approach performs task inference directly within the model, is able to dynamically expand to capture new concepts over its lifetime, and incorporates additional rehearsal-based techniques to deal with catastrophic forgetting. We demonstrate the efficacy of CURL in an unsupervised learning setting with MNIST and Omniglot, where the lack of labels ensures no information is leaked about the task. Further, we demonstrate strong performance compared to prior art in an i.i.d setting, or when adapting the technique to supervised tasks such as incremental class learning.

## 1 Introduction

Humans have the impressive ability to learn many different concepts and perform different tasks in a sequential lifelong setting. For example, infants learn to interact with objects in their environment without clear specification of tasks (*task-agnostic*), in a sequential fashion without forgetting (*non-stationary*), from temporally correlated visual inputs (*non-i.i.d*), and with minimal external supervision (*unsupervised*). For a learning system such as a robot deployed in the real world, it is highly desirable to satisfy these desiderata as well. In contrast, learning algorithms often require input samples to be shuffled in order to satisfy the i.i.d. assumption, and have been shown to perform poorly when trained on sequential data, with newer tasks or concepts overwriting older ones; a phenomenon known as catastrophic forgetting (McCloskey & Cohen, 1989; Goodfellow et al., 2013). As a result, there has been renewed research focus on the *continual learning* problem in recent years (e.g. Kirkpatrick et al., 2017; Nguyen et al., 2017; Zenke et al., 2017; Shin et al., 2017), with several approaches addressing catastrophic forgetting as well as backwards or forwards transfer—using the current task to improve performance on past or future tasks. However, most of these techniques have focused on a sequence of tasks in which both the identity of the task (*task label*) and boundaries between tasks are provided; moreover, they often focus on the supervised learning setting, where *class labels* for each data point are given. Thus, many of these methods fail to capture some of the aforementioned properties of real-world continual learning, with unknown task labels or poorly defined task boundaries, or when abundant class-labelled data is not available. In this paper, we propose to address the more general *unsupervised continual learning* setting (also suggested separately by Smith et al. (2019)), in which task labels and boundaries are not provided

---

[*]Correspondence to: {dushyantr, visin}@google.com

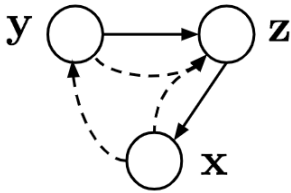

Figure 1: Graphical model for CURL. The categorical task variable $\mathbf{y}$ is used to instantiate a latent mixture-of-Gaussians $\mathbf{z}$, which is then decoded to $\mathbf{x}$.

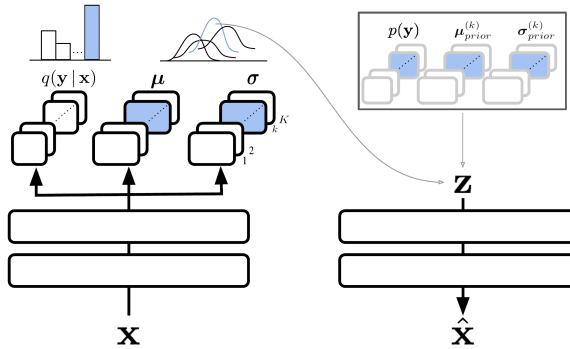

Figure 2: Diagram of the proposed approach, showing the inference procedure and architectural components used.

to the learner, and hence the focus is on unsupervised task learning. The tasks could correspond to either unsupervised representation learning, or learning skills without extrinsic reward if applied to the reinforcement learning domain. In this sense, the problem setting is "unsupervised" in two ways: in terms of the absence of task labels (or indeed well-defined tasks themselves), and in terms of the absence of external supervision such as class labels, regression targets, or external rewards. The two aspects may seem independent, but considering the unsupervised learning problem encourages solutions that aim to capture all fundamental properties of the data, which in turn might encourage, or reinforce, particular ways of addressing the task boundary problem. Hence the two aspects are connected through the type of solutions they necessitate, and it is beneficial to consider them jointly. We argue that this is an important and challenging open problem, as it enables continual learning in environments without clearly defined tasks and goals, and with minimal external supervision. Relaxing these constraints is crucial to performing lifelong learning in the real world.

Our approach, named *Continual Unsupervised Representation Learning* (CURL), learns a task-specific representation on top of a larger set of shared parameters, and deals with task ambiguity by performing task inference within the model. We endow the model with the ability to dynamically expand its capacity to capture new tasks, and suggest methods to minimise catastrophic forgetting. The model is experimentally evaluated in a variety of unsupervised settings: when tasks or classes are presented sequentially, when training data are shuffled, and with ambiguous task boundaries when transitions are continuous rather than discrete. We also demonstrate that despite focusing on unsupervised learning, the method can be trivially adapted to supervised learning while removing the reliance on task knowledge and class labels. The experiments demonstrate competitive performance with respect to previous work, with the additional ability to learn without supervision in a continual learning setting, and indicate the efficacy of the different components of the proposed method.

## 2 Model

We begin by defining the CURL model and training loss, then introduce methods to perform dynamic expansion, and propose a generative replay mechanism to combat forgetting.

### 2.1 Inference over tasks

To address the problem, we utilise the following generative model (Figure 1):

$$
\begin{aligned}
\mathbf{y} &\sim \mathrm{Cat}(\boldsymbol{\pi}), \\
\mathbf{z} &\sim \mathcal{N}(\boldsymbol{\mu}_z(\mathbf{y}), \boldsymbol{\sigma}_z^2(\mathbf{y})), \\
\mathbf{x} &\sim \mathrm{Bernoulli}(\boldsymbol{\mu}_x(\mathbf{z})),
\end{aligned}
\tag{1}
$$

with the joint probability factorising as $p(\mathbf{x}, \mathbf{y}, \mathbf{z}) = p(\mathbf{y})p(\mathbf{z} \,|\, \mathbf{y})p(\mathbf{x} \,|\, \mathbf{z})$. Here, the categorical variable $\mathbf{y}$ indicates the current task, which is then used to instantiate the task-specific Gaussian parameters for latent variable $\mathbf{z}$, which is then decoded to produce the input $\mathbf{x}$. $p(\mathbf{y})$ is a fixed uniform prior, with component weights specified by $\boldsymbol{\pi}$. In the representation learning scenario, $\mathbf{y}$ can be interpreted as representing some discrete clusters in the data, with $\mathbf{z}$ then representing

a mixture of Gaussians which encodes both the inter- and intra-cluster variation. Posterior inference of $p(\mathbf{y}, \mathbf{z} \mid \mathbf{x})$ in this model is intractable, so we employ an approximate variational posterior $q(\mathbf{y}, \mathbf{z} \mid \mathbf{x}) = q(\mathbf{y} \mid \mathbf{x}) q(\mathbf{z} \mid \mathbf{x}, \mathbf{y})$.

Each of these components is parameterised by a neural network: the input is encoded to a shared representation, the mixture probabilities $q(\mathbf{y} \mid \mathbf{x})$ are determined by an output softmax "task inference" head, and the Gaussian parameters for $q(\mathbf{z} \mid \mathbf{x}, \mathbf{y} = k)$ are produced by the output of a component-specific latent encoding head (one for each component $k$). The component-specific prior parameters $\boldsymbol{\mu}_z(\mathbf{y})$ and $\boldsymbol{\sigma}_z(\mathbf{y})$ are parameterised as a linear layer (followed by a softplus nonlinearity for the latter) using a one-hot representation of $\mathbf{y}$ as the input. Finally, the decoder is a single network that maps from the mixture-of-Gaussians latent space $\mathbf{z}$ to the reconstruction $\hat{\mathbf{x}}$. The architecture is shown in Figure 2, where for simplicity, we denote the parameters of the $k^{th}$ Gaussian by $\{\boldsymbol{\mu}^{(k)}, \boldsymbol{\sigma}^{(k)}\}$. The loss for this model is the evidence lower bound (ELBO) given by:

$$
\begin{aligned}
\log p(\mathbf{x}) \geq \mathcal{L} &= \mathbb{E}_{q(\mathbf{y}, \mathbf{z} \mid \mathbf{x})} \left[ \log p(\mathbf{x}, \mathbf{y}, \mathbf{z}) - \log q(\mathbf{y}, \mathbf{z} \mid \mathbf{x}) \right] \\
&= \mathbb{E}_{q(\mathbf{y} \mid \mathbf{x}) q(\mathbf{z} \mid \mathbf{x}, \mathbf{y})} \left[ \log p(\mathbf{x} \mid \mathbf{z}) \right] - \mathbb{E}_{q(\mathbf{y} \mid \mathbf{x})} \left[ \mathrm{KL}(q(\mathbf{z} \mid \mathbf{x}, \mathbf{y}) \,\|\, p(\mathbf{z} \mid \mathbf{y})) \right] \quad (2) \\
&\quad - \mathrm{KL}(q(\mathbf{y} \mid \mathbf{x}) \,\|\, p(\mathbf{y}))
\end{aligned}
$$

The expectation over $q(\mathbf{y} \mid \mathbf{x})$ can be computed exactly by marginalising over the $K$ categorical options, but the expectation over $q(\mathbf{z} \mid \mathbf{x}, \mathbf{y})$ is intractable, and requires sampling. The resulting Monte Carlo approximation comprises a set of familiar terms, some of which correspond clearly to the single-component VAE (Kingma & Welling, 2013; Rezende et al., 2014):

$$
\begin{aligned}
\mathcal{L} \approx \sum_{k=1}^{K} \overbrace{q(\mathbf{y} = k \mid \mathbf{x})}^{\text{component posterior}} \Bigg[ \overbrace{\log p(\mathbf{x} \mid \widetilde{\mathbf{z}}^{(k)})}^{\text{component-wise reconstruction loss}} - \overbrace{\mathrm{KL}(q(\mathbf{z} \mid \mathbf{x}, \mathbf{y} = k) \,\|\, p(\mathbf{z} \mid \mathbf{y} = k))}^{\text{component-wise regulariser}} \Bigg] \\
- \underbrace{\mathrm{KL}(q(\mathbf{y} \mid \mathbf{x}) \,\|\, p(\mathbf{y}))}_{\text{Categorical regulariser}} \quad (3)
\end{aligned}
$$

where $\widetilde{\mathbf{z}}^{(k)} \sim q(\mathbf{z} \mid \mathbf{x}, \mathbf{y} = k)$ is sampled using the reparametrisation trick. Of course, this can be generalised to multiple samples in a similar fashion to the Importance-Weighted Autoencoder (IWAE) (Burda et al., 2015).

Intuitively, this loss encourages the model to reconstruct the data and perform clustering where possible. For a given data point, the model can choose to have high entropy over $q(\mathbf{y} \mid \mathbf{x})$, in which case all of the component-wise losses must be low, or assign high $q(\mathbf{y} = k \mid \mathbf{x})$ for some $k$, and use that component to model the datum well. By exploiting diversity in the input data, the model can learn to utilise different components for different discrete structures (such as classes) in the data.

## 2.2 Component-constrained learning

While our main aim is to operate in an unsupervised setting, there may be cases in which one may wish to train a specific component, or when labels can be generated in a self-supervised fashion. In such cases where labels $\mathbf{y}_{obs}$ are available, we can use a supervised loss, adapted from Eqn. 3:

$$
\begin{aligned}
\mathcal{L}_{sup} &= \log p(\mathbf{x} \mid \widetilde{\mathbf{z}}^{(\mathbf{y}_{obs})}, \mathbf{y} = \mathbf{y}_{obs}) - \mathrm{KL}(q(\mathbf{z} \mid \mathbf{x}, \mathbf{y} = \mathbf{y}_{obs}) \,\|\, p(\mathbf{z} \mid \mathbf{y} = \mathbf{y}_{obs})) \\
&\quad + \log q(\mathbf{y} = \mathbf{y}_{obs} \mid \mathbf{x}). \quad (4)
\end{aligned}
$$

Here, instead of marginalising over $\mathbf{y}$ as in Equation 3, the component-wise ELBO (the first two terms) is computed only for the known label $\mathbf{y}_{obs}$. Furthermore, the final term in the original ELBO is replaced with a supervised cross-entropy term encouraging $q(\mathbf{y} \mid \mathbf{x})$ to match the label, which reduces to the log posterior probability of the observed label. This loss will be utilised and further discussed in Sections 2.3 and 2.4.

## 2.3 Dynamic expansion

To determine the number of mixture components, we opt for a dynamic expansion approach in which capacity is added as needed, by maintaining a small set of poorly-modelled samples and then initialising and fitting a new component to this set when it reaches a critical size. In a similar

fashion to existing techniques such as the Forget-Me-Not process (Milan et al., 2016) and Dirichlet process (Teh, 2010), we rely on a threshold to determine when to instantiate a new component. More concretely, we denote a subset of parameters $\theta^{(k)} = \{\theta_{q_y}^{(k)}, \theta_{q_z}^{(k)}, \theta_{p_z}^{(k)}\}$ corresponding to the parameters unique to each component $k$ (i.e. the $k^{\text{th}}$ softmax output in $q(\mathbf{y} \,|\, \mathbf{x})$ and the $k^{\text{th}}$ Gaussian component in $p(\mathbf{z} \,|\, \mathbf{y})$ and $q(\mathbf{z} \,|\, \mathbf{y}, \mathbf{x})$). During training, any sample with a log-likelihood less than a threshold $c_{new}$ is added to set $\mathcal{D}_{new}$ (where the log-likelihood is approximated by the ELBO). Then, when the set $\mathcal{D}_{new}$ reaches size $N_{new}$, we initialise the parameters of the new component to the current component $k^*$ that has greatest probability over $\mathcal{D}_{new}$:

$$\theta^{(K+1)} = \theta^{(k^*)}, \qquad k^* = \underset{k \in \{1,2,...,K\}}{\arg\max} \sum_{\mathbf{x} \in \mathcal{D}_{new}} q(\mathbf{y} = k \,|\, \mathbf{x}). \tag{5}$$

The new component is then tuned to $\mathcal{D}_{new}$, by performing a small fixed number of iterations of gradient descent on all parameters $\theta$, using the component-constrained ELBO (Eqn. 4) with label $K + 1$.

Intuitively, this process encourages forward transfer, by initialising new concepts to the "closest" existing concept learned by the model and then finetuning to a small number of instances. The additional capacity used for each expansion is only in the top-most layer of the encoder, with $\sim 10^4$ parameters, compared to $\sim 2.5 \times 10^6$ for the rest of the shared model. That is, while dynamic expansion incorporates a new high-level concept, the underlying low-level representations in the encoder, and the entire decoder, are both shared among all tasks.

## 2.4 Combatting forgetting via mixture generative replay

A shared low-level representation can mean that learning new tasks interferes with previous ones, leading to forgetting. One relevant technique to address this is Deep Generative Replay (DGR) (Shin et al., 2017), in which samples from a learned generative model are reused in learning. We propose to adapt and extend DGR to the mixture setting to perform unsupervised learning without forgetting. In contrast to the original DGR work, our approach is inherently generative, such that a generative replay-based approach can be incorporated holistically into the framework at minimal cost. We note that many other existing methods (e.g., Kirkpatrick et al. (2017)) could straightforwardly be adapted to our approach, but our experiments demonstrated generative replay to be simple and effective.

To be more precise, during training, the model alternates between batches of real data, with samples $\mathbf{x}_{data} \sim \mathcal{D}$ drawn from the current training distribution, and generated data, with samples $\mathbf{x}_{gen}$ produced by the previous snapshot of the model (with parameters $\theta_{prev}$):

$$\mathbf{y}_{gen} \sim \boldsymbol{\pi}(\mathbf{y}), \; \mathbf{z}_{gen} \sim p_{\theta_{prev}}(\mathbf{z} \,|\, \mathbf{y}_{gen}), \; \mathbf{x}_{gen} \sim p_{\theta_{prev}}(\mathbf{x} \,|\, \mathbf{z}_{gen}), \tag{6}$$

where $\boldsymbol{\pi}$ represents a choice of prior distribution for the categorical $\mathbf{y}$. While the uniform prior $p(\mathbf{y})$ is a natural choice, this fails to consider the degree to which different components are used, and can therefore result in poor sample quality. To address this, the model maintains a count over components by accumulating the mean of posterior $q(\mathbf{y} \,|\, \mathbf{x})$ over all previous timesteps, thereby favouring the components that have been used the most. We refer to this process as *mixture generative replay* (MGR).

While MGR ensures tasks or concepts that have been previously learned by the model are reused for learning, it places no constraint on which components are used to model them. Given that each generated datum $\mathbf{x}_{gen}$ is conditioned on a sampled $\mathbf{y}_{gen}$, we can use $\mathbf{y}_{gen}$ as a self-supervised learning signal and encourage mixture components to remain consistent with respect to the model snapshot, by using the component-constrained loss from Eqn. 4.

The only remaining question is when to update the previous model snapshot $\theta_{prev}$. For this, we explore two cases, with snapshots taken at periodic *fixed* intervals, or immediately before performing *dynamic* expansion. The intuition behind the latter is that dynamic expansion is performed when there is a sufficient shift in the input distribution, and consolidating previously learned information is beneficial prior to adding a newly observed concept. This is also advantageous as it eliminates the additional snapshot period hyperparameter.

# 3 Related Work

**Generative models**  A number of related approaches aim to learn a discriminative latent space using generative models. Building on the original VAE (Kingma & Welling, 2013), Nalisnick et al. (2016) utilise a latent mixture of Gaussians, aiming to capture class structure in an unsupervised fashion, and propose a Bayesian non-parametric prior, further developed in (Nalisnick & Smyth, 2017). Similarly, Joo et al. (2019) suggest a Dirichlet posterior in latent space to avoid some of the previously observed component-collapsing phenomena. Lastly, Jiang et al. (2017) propose Variational Deep Embedding (VaDE) focused on the goal of clustering in an i.i.d setting. While VaDE has the same generative process as CURL, it assumes a mean-field approximation, with $\mathbf{y}$ and $\mathbf{z}$ conditionally independent given the input. In the case of CURL, conditioning $\mathbf{z}$ on $\mathbf{y}$ ensures we can adequately capture the inter- and intra- class uncertainty of a sample within the same structured latent space $\mathbf{z}$.

**Continual learning**  A large body of work has addressed the continual learning problem (Parisi et al., 2019). *Regularisation-based* methods minimise changes to parameters that are crucial for earlier tasks, with some parameter-wise weight to measure importance (Kirkpatrick et al., 2017; Nguyen et al., 2017; Zenke et al., 2017; Aljundi et al., 2018; Schwarz et al., 2018). Related techniques seek to ensure the performance on previous data does not decrease, by employing constrained optimisation (Lopez-Paz et al., 2017; Chaudhry et al., 2018) or distilling the information from old models or tasks (Li & Hoiem, 2018). In a similar vein, other methods encourage new tasks to utilise previously unused parameters, either by finding "free" linear parameter subspaces (He & Jaeger, 2018); learning an attention mask over parameters (Serra et al., 2018); or using an agent to find new activation paths through a network (Fernando et al., 2017). *Expansion-based* models dynamically increase capacity to allow for additional tasks (Rusu et al., 2016; Yoon et al., 2017; Draelos et al., 2017), and optionally prune the network to constrain capacity (Zhou et al., 2012; Golkar et al., 2019). Another popular approach is that of *rehearsal-based* methods (Robins, 1995), where the data distribution from earlier tasks is captured by samples from a generative model trained concurrently (Shin et al., 2017; van de Ven & Tolias, 2018; Ostapenko et al., 2018). Farquhar & Gal (2018) combine such methods with regularisation-based approaches under a Bayesian interpretation. Alternatively, Rebuffi et al. (2017) learn class-specific exemplars instead of a generative model. However, these methods usually require task identities, rely on well-defined task boundaries, and are often evaluated on a sequence of supervised learning tasks.

**Task-agnostic continual learning**  Some recent work has investigated continual learning without task labels or boundaries. Hsu et al. (2018) and van de Ven & Tolias (2019) identify the scenarios of incremental task, domain, and class learning; which operate without task labels in the latter cases, but all focus on supervised learning tasks. Aljundi et al. (2019) propose a task-free approach to continual learning related to ours, which mitigates forgetting using the regularisation-based Memory Aware Synapses (MAS) approach (Aljundi et al., 2018), maintains a hard example buffer to better estimate the regularisation weights, and detects when to update these weights (usually performed at known task boundaries in previous work). Zeno et al. (2018) propose a Bayesian task-agnostic learning update rule for the mean and variance of each parameter, and demonstrate its ability to handle ambiguous task boundaries. However, it is only applied to supervised tasks, and can exploit the "label" trick, inferring the task based on the class label. In contrast, Achille et al. (2018) address the problem of unsupervised learning in a sequential setting by learning a disentangled latent space with task-specific attention masks, but the main focus is on learning across datasets, and the method relies on abrupt shifts in data distribution between datasets. Our approach builds upon this existing body of work, addressing the full unsupervised continual learning problem, where task labels and boundaries are unknown, and the tasks themselves are without class supervision. We argue that addressing this problem is critical in order to tackle continual learning in challenging, real-world scenarios.

# 4 Experiments

In the following sections, we empirically evaluate a) whether our method learns a meaningful class-discriminable latent space in the unsupervised sequential learning setting, without forgetting, even when task boundaries are unclear; b) the importance of the dynamic expansion and generative replay techniques to performance; and c) how CURL performs on external benchmarks when

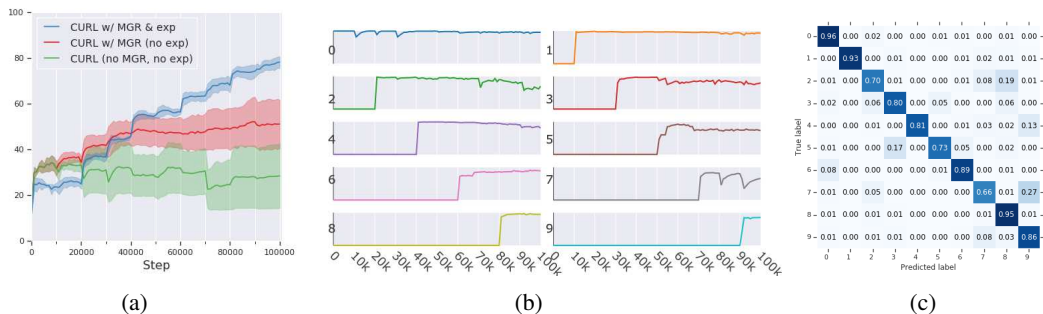

Figure 3: a) Cluster accuracy for CURL variants on MNIST, measuring the contribution of mixture generative replay ("MGR") and dynamic expansion ("exp"); b) Accuracy per class, over time; c) Class confusion matrix at the end of learning, for CURL w/ MGR & exp.

trained i.i.d or adapted to learn in a supervised fashion. Code for all experiments can be found at https://github.com/deepmind/deepmind-research/.

## 4.1 Evaluation settings and datasets

One desired outcome of our approach is the ability to learn class-discriminative latent representations from non-stationary input data. We evaluate this using cluster accuracy (the accuracy obtained when assigning each mixture component to its most represented class), and with the accuracy of a k-Nearest Neighbours (k-NN) classifier in latent space. The former measures the amount of class-relevant information encoded into the categorical variable $\mathbf{y}$, while the latter measures the discriminability of the entire latent space without imposing structure (such as a linear boundary).

For the evaluation we extensively utilise the MNIST (LeCun et al., 2010) and Omniglot (Lake et al., 2011) datasets, and further information can be found in Appendix B. We investigate a number of different evaluation settings: *i.i.d*, where the model sees shuffled training data; *sequential*, where the model sees classes sequentially; and *continuous drift*, similar to the sequential case, but with classes gradually introduced by slowly increasing the number of samples from the new class within a batch.

## 4.2 Continual class-discriminative representation learning

We begin by analysing our approach, and follow this with evaluation on external benchmarks in later sections. First, we measure the ability to perform class-discriminative representation learning in the *sequential* setting on MNIST, where each of the classes is observed for 10000 training steps (further experimental details can be found in Appendix C.1). Figure 4a shows the cluster accuracy for a number of variants of CURL. We observe the importance of both dynamic expansion and mixture generative replay (MGR) to learn a coherent representation without forgetting. Figure 4b shows the class-wise accuracies during training, for the model with MGR and expansion. Interestingly, while many existing continual learning approaches appear to forget earlier classes (see e.g. Nguyen et al. (2017)), these classes are well modelled by CURL, and the confusion is more observed between similar classes (such as 3s and 5s; or 7s and 9s). Indeed, this is reflected in the class-confusion matrix after training (Figure 4c). This implies the model adequately addresses catastrophic forgetting, but could improve in terms of plasticity, i.e., learning new concepts. Further analysis can be found in Appendix A.1, showing generated samples; and Appendix A.2, analysing the dynamic expansion buffers.

## 4.3 Ablation studies

Next, we perform an ablation study to gauge the impact of the expansion threshold for continual learning, in terms of cluster accuracy and number of components used, as shown in Figure 3. As the threshold value is increased, samples are more frequently stored into the "poorly-modelled" buffer, and the model expands more aggressively throughout learning. Consequently, for sequential learning, the number of components ranges from 12 to 71, the cluster accuracy varies up to a maximum of 84%,

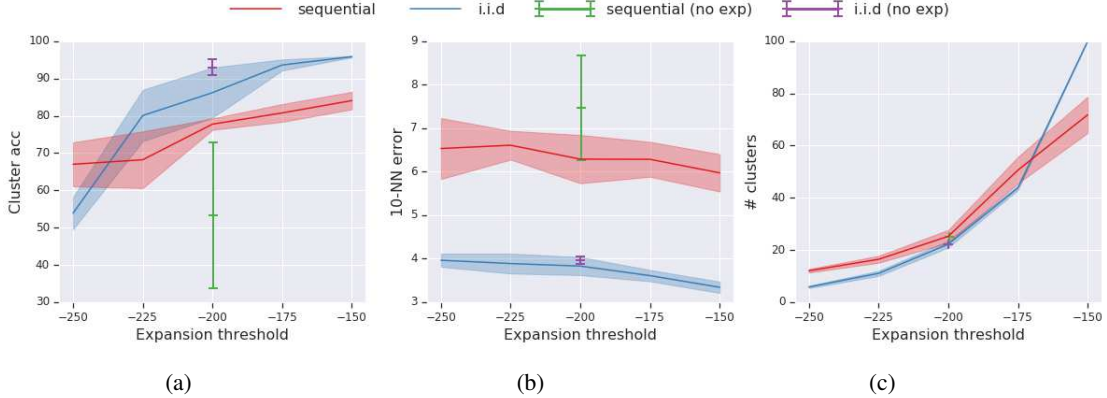

Figure 4: Ablation study for dynamic expansion on MNIST, showing (a) cluster accuracy; (b) 10-NN error; and (c) number of components used; when varying the expansion threshold $c_{exp}$. For comparison, we also show the performance without expansion ("no exp"), but using the same number of components as in the $c_{exp} = -200$ case.

| Benchmark | MNIST | | | Omniglot | | |
|---|---|---|---|---|---|---|
| Scenario | # clusters | Cluster acc (%) ↑ | 10-NN error (%) ↓ | # clusters | Cluster acc (%) ↑ | 10-NN error (%) ↓ |
| MGR (fixed, T) | $25.20_{\pm2.23}$ | $77.74_{\pm1.37}$ | $6.29_{\pm0.50}$ | $101.20_{\pm8.45}$ | $13.21_{\pm0.53}$ | $76.34_{\pm1.10}$ |
| MGR (fixed, 0.1T) | $37.60_{\pm2.15}$ | $49.14_{\pm3.95}$ | $14.95_{\pm0.73}$ | $131.60_{\pm15.74}$ | $12.13_{\pm1.54}$ | $81.21_{\pm2.06}$ |
| MGR (dyn) | $35.20_{\pm2.79}$ | $57.76_{\pm1.43}$ | $12.08_{\pm1.19}$ | $127.20_{\pm16.67}$ | $12.74_{\pm0.60}$ | $80.56_{\pm1.39}$ |
| SMGR (fixed, T) | $28.20_{\pm0.40}$ | $69.27_{\pm1.46}$ | $7.50_{\pm0.57}$ | $105.20_{\pm5.56}$ | $11.32_{\pm0.52}$ | $76.62_{\pm1.49}$ |
| SMGR (fixed, 0.1T) | $39.80_{\pm6.05}$ | $48.18_{\pm1.72}$ | $15.48_{\pm0.81}$ | $137.40_{\pm9.75}$ | $9.01_{\pm2.17}$ | $85.73_{\pm5.84}$ |
| SMGR (dyn) | $36.00_{\pm2.45}$ | $53.97_{\pm3.52}$ | $11.72_{\pm1.16}$ | $152.20_{\pm25.02}$ | $10.48_{\pm1.10}$ | $84.44_{\pm4.10}$ |
| CURL (no MGR) | $55.80_{\pm1.94}$ | $45.35_{\pm1.50}$ | $17.46_{\pm1.25}$ | $189.60_{\pm9.75}$ | $13.36_{\pm1.06}$ | $81.91_{\pm1.36}$ |

Table 1: Ablation study for mixture generative replay (MGR and SMGR), indicating the performance and number of components used. All variants perform dynamic expansion

and the $k$-NN error also marginally decreases over this range. Furthermore, without any dynamic expansion, the result is significantly poorer at $51\%$ accuracy, and when discovering the same number of components with dynamic expansion (25, obtained with an expansion threshold of $-200$), the equivalent performance is at $77\%$. Thus, the dynamic expansion threshold conveniently provides a tuning parameter to perform capacity estimation, trading off cluster accuracy with the memory cost of using additional components in the latent mixture. Interestingly, if we perform the same analysis for i.i.d. data (also in Figure 3), we observe a similar trade-off; though the final performance is slightly poorer than when starting with an equivalent, fixed number of mixture components (22).

We also further analyse mixture generative replay (MGR) with an ablation study in Table 1. We evaluate standard and self-supervised MGR (SMGR), and compare between the case where snapshots are taken on expansion (i.e., no task information is needed), or at fixed intervals (either at $T$, the duration of training on each class, or $0.1T$, ten times more frequently). Intuitively, the period is important as it determines how quickly a shifting data distribution is consolidated into the model: if too short, the generated data will drift with the model, leading to forgetting. The results in Table 1 point to a number of interesting observations. First, both MGR and SMGR are sensitive to the fixed snapshot period: the performance is unsurprisingly optimal when snapshots are taken as the training class changes, but drops significantly when performed more frequently, and also uses a greater number of clusters in the process. Second, by taking snapshots before dynamic expansion instead, this performance can largely be recovered, and without any knowledge of the task boundaries. Third, perhaps surprisingly, SMGR harms performance compared to MGR. This may be due to the fact that mixture components already tend to be consistent in latent space throughout learning, and SMGR may be reducing plasticity; further analysis can be found in Appendix A.3. Lastly, we can also observe the benefits of MGR, with the MNIST case exhibiting far poorer performance and utilising many more components in the process. Interestingly, the Omniglot case without MGR performs well, but at the cost of significantly more components: expansion itself is able to partly address catastrophic forgetting by effectively oversegmenting the data.

| Benchmark | MNIST | | | Omniglot | | |
|---|---|---|---|---|---|---|
| Scenario | # clusters | Cluster acc (%) ↑ | 10-NN error (%) ↓ | # clusters | Cluster acc (%) ↑ | 10-NN error (%) ↓ |
| Seq. w/ MGR (fixed) | $25.20_{\pm 2.23}$ | $77.74_{\pm 1.37}$ | $6.29_{\pm 0.50}$ | $101.20_{\pm 8.45}$ | $13.21_{\pm 0.53}$ | $76.34_{\pm 1.10}$ |
| Seq. w/ MGR (dyn) | $35.20_{\pm 2.79}$ | $57.76_{\pm 1.43}$ | $12.08_{\pm 1.19}$ | $127.20_{\pm 16.67}$ | $12.74_{\pm 0.60}$ | $80.56_{\pm 1.39}$ |
| Cont. w/ MGR (fixed) | $44.60_{\pm 2.65}$ | $79.38_{\pm 4.26}$ | $6.56_{\pm 0.42}$ | $111.40_{\pm 3.77}$ | $13.17_{\pm 0.37}$ | $75.80_{\pm 1.19}$ |
| Cont. w/ MGR (dyn) | $50.40_{\pm 1.85}$ | $64.93_{\pm 2.09}$ | $9.88_{\pm 1.43}$ | $129.20_{\pm 2.14}$ | $13.54_{\pm 0.35}$ | $78.78_{\pm 0.39}$ |

Table 2: Performance comparison between the sequential learning setting (with discrete changes in class), versus the continuous drift setting (with class ratios gradually changing).

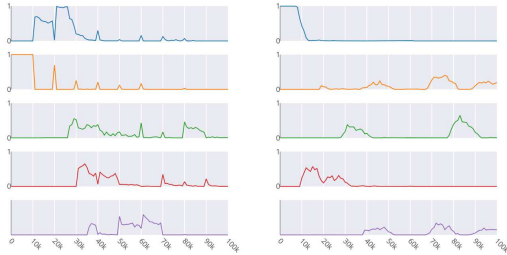

Figure 5: Mixture probabilities of the 5 components used most throughout training, with discrete class changes (left), and with continuous class drift (right).

| Benchmark | SplitMNIST | |
|---|---|---|
| Evaluation | Incr. Task | Incr. Class |
| EWC | $98.64_{\pm 0.22}$ | $20.01_{\pm 0.06}$ |
| SI | $99.09_{\pm 0.15}$ | $19.99_{\pm 0.06}$ |
| MAS | $99.22_{\pm 0.21}$ | $19.52_{\pm 0.29}$ |
| LwF | $\mathbf{99.60_{\pm 0.03}}$ | $24.17_{\pm 0.33}$ |
| GEM | $98.42_{\pm 0.10}$ | $92.20_{\pm 0.12}$ |
| DGR | $99.50_{\pm 0.03}$ | $91.24_{\pm 0.33}$ |
| iCARL | - | $\mathbf{94.57_{\pm 0.11}}$ |
| CURL | $99.10_{\pm 0.06}$ | $92.59_{\pm 0.66}$ |

Table 3: Supervised learning benchmark on splitMNIST, for incremental task and incremental class learning. [2]

## 4.4 Learning with poorly-defined task boundaries

Next, we evaluate CURL in the *continuous drift* setting, and compare to the standard sequential setting. The overall performance on MNIST and Omniglot is shown in Table 2, using MGR with either fixed or dynamic snapshots. We observe that despite having unclear task boundaries, where classes are gradually introduced, the continuous case generally exhibits better performance than the case with well-defined task boundaries. We also closely investigate the mixture component dynamics during learning, by obtaining the top 5 components (most used over the course of learning) and plotting their posterior probabilities over time (Figure 5). From the discrete task-change domain (left), we observe that probabilities change sharply with the hard task boundaries (every 10000 steps); and many mixture components are quite sparsely activated, modelling either a single class, or a few classes. Some of the mixture components also observe "echoes", where the sharp change to a new class in the data distribution activates the component temporarily before dynamic expansion is performed. In the continuous drift case (right of Figure 5), the mixture probabilities exhibit similar behaviours, but are much smoother in response to the gradually changing data distribution. Further, without a sharp distributional shift, the "echoes" are not observed.

## 4.5 External benchmarks

**Supervised continual learning** While focused on task-agnostic continual learning in unsupervised settings, CURL can also be trivially adapted to supervised tasks simply by training with the supervised loss in Eqn. 4. We evaluate on the split MNIST benchmark, where the data are split into five tasks, each classifying between two classes, and the model is trained on each task sequentially. If we evaluate the overall accuracy after training, this is called *incremental class learning*; and if we provide the model with the appropriate task label and evaluate the binary classification accuracy for each task, this is *incremental task learning* (Hsu et al., 2018; van de Ven & Tolias, 2019). Experimental details can be found in Appendix C.2. The results in Table 3 demonstrate that the proposed unsupervised approach can easily and effectively be adapted to supervised tasks, achieving competitive results for both scenarios. While all methods perform quite well on incremental task learning, CURL is outperformed only by iCARL (Rebuffi et al., 2017) on incremental class learning, which was specifically proposed for this task. Interestingly, the result is also better than DGR, suggesting that by holistically incorporating the generative process and classifier into the same model, and focusing on the broader unsupervised, task-agnostic perspective, CURL is still effective in the supervised domain.

| Benchmark | MNIST ($n_z = 50$) | | | Omniglot ($n_z = 100$) | | |
|---|---|---|---|---|---|---|
| Evaluation | 3-NN error | 5-NN error | 10-NN error | 3-NN error | 5-NN error | 10-NN error |
| VAE[3] | $27.16_{\pm 0.48}$ | $20.20_{\pm 0.93}$ | $14.89_{\pm 0.40}$ | $92.34_{\pm 0.25}$ | $91.21_{\pm 0.18}$ | $88.79_{\pm 0.35}$ |
| SBVAE[3] | $10.01_{\pm 0.52}$ | $9.58_{\pm 0.47}$ | $9.39_{\pm 0.54}$ | $86.90_{\pm 0.82}$ | $85.10_{\pm 0.89}$ | $82.96_{\pm 0.64}$ |
| DirVAE[3] | $5.98_{\pm 0.06}$ | $5.29_{\pm 0.06}$ | $5.06_{\pm 0.06}$ | $\mathbf{76.55_{\pm 0.23}}$ | $\mathbf{73.81_{\pm 0.29}}$ | $\mathbf{70.95_{\pm 0.29}}$ |
| CURL (i.i.d) | $4.40_{\pm 0.34}$ | $4.22_{\pm 0.28}$ | $4.23_{\pm 0.30}$ | $78.18_{\pm 0.47}$ | $75.41_{\pm 0.34}$ | $72.51_{\pm 0.46}$ |
| VaDE (bigger net) | $\mathbf{2.20}$ | $\mathbf{2.14}$ | $\mathbf{2.22}$ | - | - | - |
| CURL w/ MGR (seq) | $4.58_{\pm 0.26}$ | $4.35_{\pm 0.32}$ | $4.50_{\pm 0.34}$ | $83.95_{\pm 0.72}$ | $81.56_{\pm 0.75}$ | $78.80_{\pm 0.74}$ |
| Raw pixels[3] | 3.00 | 3.21 | 3.44 | 69.94 | 69.41 | 70.10 |

Table 4: Unsupervised learning benchmark comparison with sampled latents. We compare with a number of approaches trained i.i.d, as well as CURL trained in the sequential setting.

**Unsupervised i.i.d learning**   We also demonstrate the ability of the underlying model to learn in a more traditional setting with the entire dataset shuffled, and compare with existing work in clustering and representation learning: the VAE (Kingma & Welling, 2013), DirichletVAE (Joo et al., 2019), SBVAE (Nalisnick & Smyth, 2017), and VaDE (Jiang et al., 2017). We utilise the same architecture and hyperparameter settings as in Joo et al. (2019) for consistency, with latent spaces of dimension 50 and 100 for MNIST and Omniglot respectively; and full details of the experimental setup can be found in Appendix C.3. We note that the $k$-NN error values are much better here than in Section 4.3; this is due to a higher dimensional latent space and hence they cannot be directly compared (see Appendix A.4).

The uppermost group in Table 4 show the results on i.i.d MNIST and Omniglot. The CURL generative model trained i.i.d (without MGR, and with dynamic expansion) is competitive with the state-of-the-art on MNIST (bettered only by VaDE, which incorporates a larger architecture) and Omniglot (bettered only by DirVAE). While not the main focus of this paper, this demonstrates the ability of the proposed generative model to learn a structured, discriminable latent space, even in more standard learning settings with shuffled data. Table 4 also shows the performance of CURL trained in the *sequential* setting. We observe that, despite learning from sequential data, these results are competitive with the state-of-the-art approaches that operate on i.i.d. data.

## 5   Conclusions

In this work, we introduced an approach to address the unsupervised continual learning problem, in which task labels and boundaries are unknown, and the tasks themselves lack class labels or other external supervision. Our approach, named CURL, performs task inference via a mixture-of-Gaussians latent space, and uses dynamic expansion and mixture generative replay (MGR) to instantiate new concepts and minimise catastrophic forgetting. Experiments on MNIST and Omniglot showed that CURL was able to learn meaningful class-discriminative representations without forgetting in a sequential class setting (even with poorly defined task boundaries). External benchmarks also demonstrated the method to be competitive with respect to previous work when adapted to unsupervised learning from i.i.d data, and to supervised incremental class learning. Future directions will investigate additional techniques to alleviate forgetting, and the extension to the reinforcement learning domain.

## Footnotes

[2]Performances of existing approaches are taken from studies by Hsu et al. (2018) and van de Ven & Tolias (2019), using the better of the two.

[3] Performance numbers are obtained from Joo et al. (2019), with consistent architectures and hyperparameters.

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
