[Supplementary Material]

# Appendix

## A   Additional experiments

### A.1   Generated samples

The primary aim of this paper is to learn a meaningful class-discriminative representation of the data. We use mixture generative replay to contrast catastrophic forgetting, hence sample quality is not our main interest. Nonetheless, we are interested in checking whether the model is able to capture the variety of the data and what level of sample quality is sufficient to retain what has been learned. We illustrate samples generated after sequentially learning on each class, in Figure 6. We observe from the later samples (bottom rows) that classes that are observed early on are still preserved within the model. Interestingly, while most of the samples are clear, some indicate degraded but still identifiable versions of previous symbols (such as some of the zeros after other classes are introduced). Given that the learned latent representations are still class-discriminative, we hypothesise that perhaps merely capturing the "essence" of a particular class may be more useful for representation learning than striving for a pixel-perfect reconstruction. We leave a thorough analysis of this idea to future work.

### A.2   Data buffers for dynamic expansion

We can also observe samples from the data buffers used for each dynamic expansion, to understand when the model chooses to expand capacity (Figure 7). We note that some buffers holds samples from multiple classes: this occurs when there are few outliers from a single class, insufficient to initialise a new component on their own. Nonetheless, we observe an intuitive trend over most of the buffers: in many cases for a class, the first expansion denotes the change in distribution that corresponds to the introduction of the class; and the second expansion responds to outlying or challenging examples from the class. This can be observed, for instance, for the initialisation of a component for "twos" and then for "curly twos", or for "threes" and "challenging threes". This highlights the ability of the approach to incorporate new components to account for both data distributional shift and hard example modelling.

Figure 6: Model-generated samples, with each row corresponding to observing the next class in the sequential setting (top to bottom).

Figure 7: Samples stored in the poor data buffer before each successive expansion (top to bottom; left to right).

### A.3   Latent structure

To analyse the structure of our latent space, we encode our test set to latent space, and observe how the latent space changes during training (Figure 8). Figure 8(a) colours the points by ground truth class label (not available to the model during learning), and Figure 8(b) colours them by the most probable mixture component (i.e. $\arg\max_k q(\mathbf{y} = k \,|\, \mathbf{x})$). Each subplot represents a particular step during training, and only points from classes seen so far are used in the plot. We observe from Figure 8(a) that as classes are incrementally introduced, they occupy a relatively disjoint region of latent space, and are consistent over time. That is, the introduction of new classes does not appear to catastrophically interfere with the learned representations of previous ones.

We also observe similar properties in Figure 8(b), with the clusters covering individual classes with reasonable accuracy, and maintaining a consistent position over time. This may offer some explanation for the inefficacy of SMGR - there is little value to incorporating the self-supervised loss

for component consistency given that the mixture components are already consistent in which classes they model.

(a)

(b)

Figure 8: t-SNE plots showing the evolution of the latent space during learning, with points coloured by (a) class label, and (b) mixture component.

## A.4 Discussion on k-NN accuracy measure

We exploit the $k$-NN accuracy in our experiments to measure the class-discriminability of latent space, without imposing any specific parametric structure in terms of the boundary between classes. As a simple non-parametric approach, it serves this purpose, and has also been used extensively in previous evaluations as a result (Nalisnick & Smyth, 2017; Joo et al., 2019). However, there are some interesting properties that are worth considering. Firstly, as demonstrated by Table 5, the measure is highly dependent on dimensionality, making it difficult to compare across different latent space sizes. In our experiments, all architectural aspects, including latent size, are kept fixed for an experiment to account for this. Secondly, simple baselines like classifying on raw pixels perform surprisingly well (e.g., approximately 3% k-NN error on MNIST); this is due in part to the much larger dimensionality than the latent spaces used for evaluation, but we postulate that this is also likely due to the image statistics of the datasets used: given MNIST and Omniglot have many low-variance black pixels and a relatively small amount of intra-class variance within the centre pixels, the raw pixels are themselves discriminative. Given the inability to directly compare between different dimensionalities, this simple baseline is only provided for context.

| | \multicolumn{8}{c}{Latent space dimension ($n_z$)} | | | | | | | |
|---|---|---|---|---|---|---|---|---|
| | 8 | 16 | 32 | 64 | 128 | 256 | 784 | 1568 |
| 10-NN error | $58.75_{\pm 2.94}$ | $42.12_{\pm 1.76}$ | $28.20_{\pm 1.41}$ | $18.76_{\pm 0.58}$ | $14.56_{\pm 0.50}$ | $12.68_{\pm 0.64}$ | $11.14_{\pm 0.42}$ | $11.09_{\pm 0.41}$ |

Table 5: Test error on MNIST of a 10-NN classifier in latent space, for a randomly initialised model with different latent dimensions.

## B   Datasets

The MNIST dataset comprises handwritten samples of ten digits, split into $50,000$ training samples, $10,000$ validation samples, and $10,000$ test samples. The Omniglot dataset comprises 20 samples from each of 1623 characters, grouped into 50 different alphabets. While multiple splits are possible, we utilise a common method from previous work, with 15 samples from each character in the training set, and 5 in the test set. In this case, we use the 50 alphabets as the class labels for evaluation. For all experimental runs, we train models for $10^5$ training steps. In the sequential cases, this is equally divided between all classes, resulting in $10^4$ and $2 \times 10^3$ steps per class for MNIST and Omniglot, respectively.

## C   Experimental setup

We used a single main setup for most of the experiments, with the exception of the external benchmarks, which are described separately in the following sections. For all experiments, we train models for $10^5$ steps and report / plot the mean and standard deviation over 5 different random seeds. Each random seed for an experiment is trained using a single Tesla V100 GPU. Each run takes approximately 1 hour for the supervised SplitMNIST scenario, 3 hours for the main unsupervised MNIST sequential run, and 10 hours for Omniglot. The main bottleneck in run time is marginalising over all components, so this could be optimised in future implementations.

### C.1   Main setup

For both MNIST and Omniglot, we employed an MLP encoder with layer sizes of $\{1200, 600, 300, 150\}$ to form the shared representation; followed by a softmax layer for **y**, with a maximum capacity of $K = 100$ components; and a linear layer with 64 output dimensions, for each component, to output the posterior parameters of the corresponding 32-dimensional latent Gaussian. Half of the output dimensions were used directly for the mean, while the other half were passed through a softplus activation to produce the variance. The decoder consisted of two fully connected layers of 500 dimensions, followed by a Bernoulli output layer for the reconstruction. Both encoder and decoder networks employed ReLU nonlinearities, and training was performed using the ADAM optimiser with learning rate $10^{-3}$.

This architecture was obtained by performing a small hyperparameter sweep, also considering an encoder with two 500-dimensional fully-connected layers, and a decoder with layer sizes $\{150, 300, 600, 1200\}$, but we observed only small differences in performance.

For dynamic expansion, we set the threshold for the log-likelihood (approximated by the ELBO) at $c_{new} = -200$, and anything below this value is considered a poorly-explained sample and added to the buffer. This was obtained after an ablation over values of $\{-150, -175, -200, -225, -250\}$, where this parameter can be use to directly modulate the expansion rate, and hence the capacity of the model (i.e. it is a way to automatically estimate the required number of components K). A fixed consolidation period of 100 steps was used after each expansion before the model was re-eligible for expansion: this ensures that the model is able to learn from the data and fit a new component, and only flag poorly defined samples once learning has matured. For all hyperparameter sweeps, model selection was performed using the validation set.

### C.2   Supervised continual learning benchmark

For this external comparison, we employ the same hyperparameters as those reported in the previous work (Hsu et al., 2018; van de Ven & Tolias, 2019). The encoder comprised two fully-connected

ReLU layers with 400 dimensions, and we use a 100-dimensional latent space with a capacity of $K = 10$ components (matching the number of labels). The decoder also comprises two fully-connected ReLU layer with 400 dimensions, and Bernoulli outputs. Dynamic expansion is performed in a supervised fashion (i.e. expanding when new class labels are introduced), and MGR is also used, with snapshotting at the end of each task.

## C.3 Unsupervised i.i.d learning benchmark

For the external comparison, we employ the same hyperparameters as those reported in the previous work (Joo et al., 2019; Nalisnick & Smyth, 2017). For MNIST, the encoder comprised two fully-connected ReLU layers with 500 dimensions, and a 50-dimensional latent space. The decoder comprised a single fully-connected ReLU layer with 500 dimensions, and Bernoulli outputs. For Omniglot, the same architecture was used, but with a 100-dimensional latent space. By default, we employ dynamic expansion, and use MGR for the sequential learning case, using the "dynamic" snapshot approach. Training was performed using the ADAM optimiser with learning rate $5 \times 10^{-4}$.