[Reviews · NeurIPS 2019]

Reviewer 1



I like the problem introduced in the paper and the approach taken. I have few questions and suggestion to authors: 1) “For dynamic expansion, we set the threshold for the negative log-likelihood (approximated by the ELBO) at cnew = 100”: How did authors decide cnew? Was there any tuning done to decide cnew? What is the effect of low cnew in experimental results? 2) What’s the time complexity of the algorithm? How long did it take to run all the experiments on Tesla V100 GPU? 3) Did authors try any preliminary experiments on little more complex datasets like CIFAR-100? 4) Figure 3b: Why did performance of “1” is getting worse with time but performance of “0” is not? 5) Figure 3B: Performance of “5” never seem to pick up beyond 50-60%. Is there any reason why? The submission is technically sound and claims are well supported by experimental results. The submission is clearly written and well organized. Authors address a novel problem setting with combination of old techniques. It is clear how this work differs from previous contributions and related work is adequately cited. Researchers and practitioners are likely to use the ideas presented in this paper and build on them.

Reviewer 2



============== After rebuttal ============== I thank the authors for their response, they have managed to clarify some of my concerns and overall I vote for acceptance of the paper. The authors introduce a method for continual unsupervised learning. They propose a generative categorical model, in which the latent space is modeled as a mixture of Gaussians, with a Bernoulli decoder. An expansion technique is used to include new mixture components for poorly modeled examples, and the generative model is used with previous model parameters to prevent forgetting old tasks. Their method is analysed on tasks constructed around MNIST and Omniglot, with an ablation study on the expansion and generative replay. The extension to a more standard supervised setting is also presented. Novelty and quality: The exact setting proposed in the paper, as well as the proposed model, are novel to my knowledge. Significance: The main contribution of the paper is empirical. The setting of unsupervised continual learning proposed by the authors is relevant and provides an interesting proof of concept for other tasks which can benefit from it, such as reinforcement learning. The experiments in 4.1 to 4.3 lack comparison to simple baselines, such as a hierarchical clustering technique. Since comparison to other methods is not possible, I believe this would strengthen the paper. Could the authors provide such a baseline? The fact that the method does well in supervised tasks is reassuring. Clarity: The paper is overall clear and the method is well presented. I have the following detailed comments: 1) The lack of comparison to baselines, as previously mentioned. 2) Why is the accuracy in Omniglot so low in Table 1? It is difficult to draw any conclusions if the method has such a high error on the dataset. 3) Why is only one sample \tilde(z)_k used in eq 3)? 4) I would appreciate more details to understand how eq 4) is derived, it is currently rather intuitively motivated. 5) What value is chosen for N_new? How is this selected? Moreover, is it sensible to initialise the parameters of a new cluster as in eq 5) if the new class is very different from previous classes? Did you try random initialization? 6) What exactly is p_{\theta} in eq 6)? 7) How are the training and test splits determined in the experiments, and how are the error bars computed? 8) How reproducible are Figure 4 and Figure 2b?

Reviewer 3



Originality: Unsupervised continual learning treated in this paper is challenging and quite important in the context of continual learning and lifelong learning. The proposed algorithm has several desirable properties including dynamic expansion and mixture generative replay, and can deal with both unsupervised and supervised continual learning problems. Quality: Although the authors do not provide the theoretical guarantee for the algorithm, the numerical experiments show the effectiveness of the algorithm. The authors seems honest about evaluating the proposed algorithm and other state-of-the-art algorithms as long as I see Table 3 and table 4. Clarity: This paper is totally well-written and the explanation for the proposed model is intelligible. Moreover, the authors provide adequate information about related work. Significance: The problem settings treated in this paper is quite significant in continual learning and the proposed algorithm has several interesting properties. I believe this paper inspires various ideas for readers.

[Author Response · NeurIPS 2019]

# Author response for Continual Unsupervised Representation Learning

We would like to thank the reviewers for their insightful and constructive comments, which will certainly help to strengthen the paper. We have systematically addressed each of their concerns, which we summarise below.

**Clarification on dynamic expansion, and thresholds $c_{new}$ and $N_{new}$ (R1, R2)**   $c_{new}$ is similar to the alpha parameter in a Dirichlet process, controlling how easily additional components are added to the mixture. To analyse its effect, we have added a sweep over this parameter in the experiments: with lower threshold values, the model can use more components, leading to higher accuracies at the cost of additional capacity. In terms of $N_{new}$, the model is quite insensitive: we set the value to $100$ but found the performance to be quite stable for values up to $1000$. In the case where new classes are very different (asked by R2), our intuition was that the new class is still closer to existing classes than to randomly initialised weights (mentioned by R1), and we verified empirically that our approach yields better performance than this case.

**Model details (R2) and theoretical aspects (R3)**   We only use one sample for z for simplicity/efficiency, but this can easily be extended in a similar fashion to Importance-Weighted Autoencoders - this is now mentioned in the paper. We have also added additional explanation and intuition around how Eqn. 4 (the supervised loss) is derived, and how each term corresponds to its equivalent in Eqn 3. Finally, we have also added a clarification of $\theta_{prev}$ leading up to Eqn. 6: it denotes the parameters of the previous model snapshot, specifically those of the prior $p(z|y)$ and decoder $p(x|z)$.

**More in-depth discussion on results ("why") (R1)**   Interestingly, the performance of individual classes for CURL seems more dependent on similarity between classes: for example, 5 is very similar to 3, 8, and 9, which is likely why its performance is lower than the more distinguishable 0. This is in contrast to previous continual learning work, in which forgetting is often correlated with whether classes are learned earlier or later. We have added additional discussion in Section 4.2 to make this clear, as well as further class-specific plots in the appendix.

**Experimental details (R1)**   The details around error bars and training/test splits are in Appendix C. We have modified this to clarify that the errors show the standard deviation over 5 runs. The time complexity has also been added to the Appendix: training time is around $15 - 20$ minutes for MNIST and $4 - 5$ hours for Omniglot (with additional time for analysis and evaluation on validation and test sets).

**Comparison to clustering baselines (R2)**   Unfortunately, to the best of our knowledge, [hierarchical] clustering techniques still assume i.i.d data, while in our work we investigate clustering in a non-stationary setting. In the i.i.d case, we do include simple baselines in Section 4.4, reporting kNN error with raw pixels and random network encodings.

**Low Omniglot accuracy (R2)**   The reviewer is correct that the general performance on Omniglot is quite poor - this is largely due to the more challenging task with 50 classes, but performance is still much better than random chance at $2\%$. Though the large error makes it a bit more difficult to separate different techniques, we still observe strong performance with respect to baselines in the ablation and external comparison.

**Reproducibility of Figure 4 and Figure 2b (R2)**   We believe the reviewer is referring to the class-specific analysis in Figure 3b and Figure 4. This kind of behavior is very robust/stable across multiple seeds: similar classes are confused and similar numbers of components are used for each class.

**Clarification of "Unsupervised i.i.d. learning" (R3)**   By i.i.d. learning, we refer to the setting where all classes are sampled with uniform probability from the beginning of training. We have improved its definition in Section 4.1.

**Improved experiments**

- We have managed to increase performance (generally across the board) with small architectural changes and further hyperparameter tuning. The numbers have been updated and details have been added to the appendix.
- We are running experiments with CIFAR-100 (suggested by R1) and hope to have this by the camera-ready deadline. Surprisingly, there is little past work demonstrating class-discriminative unsupervised learning with CIFAR-100, even in the i.i.d setting, so we focus on the supervised incremental CIFAR-100 benchmark.
- After correspondence with authors of related work, we found that the unsupervised i.i.d benchmark was performed with sampled latents, and redid the experiments accordingly. The analysis and conclusions drawn from the original submitted version still hold, so merely the numbers have slightly changed.
- We are currently in the process of open-sourcing our code.

[Meta-Review · NeurIPS 2019]

The task of continual unsupervised learning without task labels and boundaries is interesting, and would be important for learning in the real-world system. The novelty of the proposed method is high. The paper is well written. Experiments with larger-scale data would strengthen the paper.